# Breast and Gut Microbiota Action Mechanisms in Breast Cancer Pathogenesis and Treatment

**DOI:** 10.3390/cancers12092465

**Published:** 2020-08-31

**Authors:** Aurora Laborda-Illanes, Lidia Sanchez-Alcoholado, María Emilia Dominguez-Recio, Begoña Jimenez-Rodriguez, Rocío Lavado, Iñaki Comino-Méndez, Emilio Alba, María Isabel Queipo-Ortuño

**Affiliations:** 1Unidad de Gestión Clínica Intercentros de Oncología Médica, Hospitales Universitarios Regional y Virgen de la Victoria, Instituto de Investigación Biomédica de Málaga (IBIMA)-CIMES-UMA, 29010 Málaga, Spain; aurora.laborda@ibima.eu (A.L.-I.); 0610451838@uma.es (L.S.-A.); emilia.dominguez@ibima.eu (M.E.D.-R.); bego.jimenez@ibima.eu (B.J.-R.); rocio.lavado@ibima.eu (R.L.); inaki.comino@ibima.eu (I.C.-M.); 2Facultad de Medicina, Universidad de Málaga, 29071 Málaga, Spain

**Keywords:** breast cancer, microbiota, estrobolome, immune responds, inflammation, epigenetic modulation, anticancer therapy, prebiotics, probiotics

## Abstract

**Simple Summary:**

In this review we discuss the recent knowledge about the role of breast and gut microbiome in the pathogenesis of breast cancer. We examine the proposed mechanisms of interaction between breast tumors and the microbiome. We focus on the role of the microbiome in: (i) the development and maintenance of estrogen metabolism through bacterial beta-glucuronidase enzymes (ii) the regulation of the host´s immune system and tumor immunity by Treg lymphocyte proliferation through bacterial metabolites such as butyrate and propionate (SCFAs), (iii) the induction of chronic inflammation, (iv) the response and/or resistance to treatments and (v) the epigenetic reprogramming. Moreover, we also discuss that diet, probiotics and prebiotics could exert important anticarcinogenic effects in breast cancer that could indicate their employment as adjuvants in standard-of-care breast cancer treatments. Overall, these findings could give new insights for building up novel strategies for breast cancer prevention and treatment.

**Abstract:**

In breast cancer (BC) the employment of sequencing technologies for metagenomic analyses has allowed not only the description of the overall metagenomic landscape but also the specific microbial changes and their functional implications. Most of the available data suggest that BC is related to bacterial dysbiosis in both the gut microenvironment and breast tissue. It is hypothesized that changes in the composition and functions of several breast and gut bacterial taxa may contribute to BC development and progression through several pathways. One of the most prominent roles of gut microbiota is the regulation of steroid-hormone metabolism, such as estrogens, a component playing an important role as risk factor in BC development, especially in postmenopausal women. On the other hand, breast and gut resident microbiota are the link in the reciprocal interactions between cancer cells and their local environment, since microbiota are capable of modulating mucosal and systemic immune responses. Several in vivo and in vitro studies show remarkable evidence that diet, probiotics and prebiotics could exert important anticarcinogenic effects in BC. Moreover, gut microbiota have an important role in the metabolism of chemotherapeutic drugs and in the activity of immunogenic chemotherapies since they are a potential dominant mediator in the response to cancer therapy. Then, the microbiome impact in BC is multi-factorial, and the gut and breast tissue bacteria population could be important in regulating the local immune system, in tumor formation and progression and in therapy response and/or resistance.

## 1. Introduction

Breast cancer (BC) is one of the most common tumors in women worldwide and despite significant progress in its diagnosis and treatment, there are still more than 40,000 deaths per year [1]. Clinically, patients with BC present diseases with very different outcomes and thanks to technological advances, different molecular profiles have been described in this tumor type. To date, four major invasive breast carcinoma genetic subtypes have been identified with prognostic and therapeutic relevance, such as the luminal subtype A, presenting high expression of estrogen (ERs) and progesterone (PRs) receptors without human growth factor receptor type 2 (Her2) overexpression and low cell proliferation index; the luminal subtype B (ER PR+ and high proliferation index); a group of tumors overexpressing Her2 (ER−/PR− and Her2+), and finally the triple negative subgroup or TNBC (ER/PR− and Her2−) [2,3]. The main factors associated with increased risk in these patients are advanced age, reproductive history, personal or family history of breast disease, genetic predisposition and environmental factors. Most cases are diagnosed in localized stages when the disease is potentially curable. Overall, a BC patient´s average survival after five years is 89.2%, being the tumor stage, a crucial factor influencing progression. In this regard, survival in stage I tumors is more than 98%, however in stage III survival rate decrease to 24%. There are other factors affecting BC presentation and outcome; for example, women with invasive BC have a higher risk of contralateral BC as well as those ones with in situ ductal or lobular carcinoma, or antecedents of benign proliferative disease, who have an increased risk of BC. On top of that, high breast density is also related to BC increased risk [4,5]. On the other hand, outcome differs also considering the different BC subgroups mentioned above, being the majority of the relapses after 5 years and frequently affecting luminal subtypes while TNBCs and Her2+ cases relapse more frequently and occur much earlier. The luminal A subtype has the best prognosis and the basal-like subtype is the one with the worst prognosis.

Considering all the above, the identification of cases in very early stages both using diagnostic imaging methods, such as mammography and resonance along with molecular ultrasensitive methodologies, is paramount to assess risks and clinical impact in the disease.

## 2. Approaches to Assess Microbiome: Pros and Cons

The main culture-independent approaches for analyzing the microbiome are the identification of the 16S ribosomal RNA (rRNA) as well as the shotgun metagenomics. The employment of one or another is highly relative to the research objectives. The study of the 16S rRNA is suitable for the analysis of a large number of samples, but present a limited taxonomical and functional resolution, since they do not target the entire genomic content of a sample [6]. On the other hand, the shotgun metagenomics have increased resolution, allowing one to profile taxonomic composition and functional potential of microbial communities, as well as to discover new bacterial genes and genomes. However, this approach is more expensive and requires more complex bioinformatic analyses [7]. Additionally, contamination during sample collection, preservation (freeze-thaw cycles or storage buffers) and nucleic acid isolation where commercial kits for DNA and RNA isolation have different efficiency in lysing specific microbes, are substantially affecting the metagenomic data accuracy [8]. Importantly, the above-mentioned approaches only describe the presence of microorganisms or genes, but they are not capable of describing the active from the inactive members of a given microbiome. In this regard, RNA sequencing (RNASeq) provides genes and pathways within a microbiome. Conversely, its experimental design is highly complex and presents limitations, and it is usually not capable of capturing the entire metatranscriptome profiles due to the high diversity and relative ratios of some microbial communities as well as the short RNA half-life [9].

Other direct methodologies, such as scanning electron microscopy, can be employed to visualize microbial organization in fixed samples, but this is not the technology-of-choice to identify individual taxa in complex communities [10]. Another example is the fluorescence in situ hybridization (FISH) with rRNA-targeted oligonucleotide probes to identify, visualize and quantify the microbial community members in fixed samples. Nevertheless, it is limited by the small number of phylogenetically different target organisms simultaneously detectable [11].

Metabolomic and metaproteomic technologies that utilized mass spectrometry (MS) and nuclear magnetic resonance (NMR) spectrometry are also interesting approaches to characterize the human microbiome molecular profiles. They have been used to quantify proteins and metabolites such as vitamins, bile salts, fatty acids and polyphenols produced by the microbiome [12].

Overall, microbiome structure and functions cannot be characterized employing only one strategy. Metagenomics analyses should be complemented with metatranscriptomics, metaproteomics and metabolomics data as well as with complete clinical and dietary information to carry out comprehensive and informative microbiome research studies.

## 3. Microbiota and Breast Cancer

More than half the women who develop BC have no known risk factors [13] and only a fraction of them with genetic predisposition or exposed to known environmental risk factors develop the disease [14]. Although diet, alcohol, and radiation have been associated with increased incidence, the main risk factors identified so far are exposure to hormones, including physiological variations associated with puberty, pregnancy, menopause, optional use of hormonal contraceptives and/or hormone replacement therapies [15]. However, other factors contributing to BC appearance and development need to be identified. To be overweight or obese also represents a known risk factor for BC, especially in postmenopausal women [9], but a novel aspect influencing this disease is the human microbiome [16].

The disturbance of gut microbial communities, known as dysbiosis, has been linked not only to acute disease, but also to chronic diseases and malignancies [17,18], including cancer. For example, the role of *Helicobacter pylori* in the development of stomach adenocarcinoma [19] and the role of certain intestinal microbiota profiles in the development and progression of colorectal cancer [20].

### 3.1. Mammary Microbiota and Breast Cancer

Considering the different effects that the microbiome has in distinct organs, recent studies have focused on examining colonizing bacteria in breast tissue. In this regard, specific microbiota have been identified in breast milk [21], and several authors postulated that bacteria are capable of using the nipple to gain access to the breast ducts and create a specific microbiome in the breast. This is not surprising considering that skin and oral bacteria have access to the breast ducts through the nipple [22,23], but interestingly recent studies have suggested that their origin are the mother´s gastrointestinal tract [24].

An increasing number of studies are describing the breast microbiome in BC patients (Table 1). Among them, Xuan et al. studied the potential role of the microbiota in ER+ BC by sequencing the ribosomal RNA (rRNA) 16S in breast tumor tissue and the healthy adjacent tissue from the same patient. The authors observed that the bacterium *Methylobacterium radiotolerans* was relatively enriched in the tumor tissue, while *Sphingomonas yanoikuyae* was relatively enriched in the adjacent healthy tissue. In addition, the total bacterial DNA load was reduced in the tumor when compared with healthy adjacent breast tissue and it was inversely correlated with the presence of advanced disease, an observation with crucial implications in the diagnosis and staging of BC. Finally, they observed lower baseline expression levels of antibacterial response genes in tumor tissue versus healthy breast tissue [25]. On the other hand, Yazdi et al. identified significant differences in the presence of *Methylobacterium radiotolerance* when comparing lymph cancer node samples and normal adjacent tissues [26]. Conversely, Wang et al. observed a decrease in the relative abundance of *Methylobacterium* in invasive breast carcinoma in comparison to breast tissues from healthy women. However, cancer patients had increased levels of Gram-positive organisms including *Corynebacterium, Staphylococcus, Actinomyces* and Propionibacteriaceae [27]. According to the results from Thompson et al., Proteobacteria, Actinobacteria and Firmicutes were the most prevalent phyla in breast tissues. Proteobacteria was the more prevalent in breast tumor samples; however, Actinobacteria was predominant in normal adjacent tissue [28]. Nevertheless, Meng et al. observed an increased representation of genus *Propionicimonas* and the families Micrococcaceae, Caulobacteraceae, Rhodobacteraceae, Nocardioidaceae and Methylobacteriaceae in malignant breast tumor tissues using a Chinese cohort of patients, although it is important to consider that these results are probably affected by the ethno-specific characteristic of the set [29]. Banerjee et al. published that each BC subtype also had its own unique viral, bacterial, fungal and parasitic breast signature. In this regard, ER and HER2 positive BC subtypes showed more similar patterns than TNBC tissues. The main signatures common to all four types identified in this study were of the Proteobacteria, although Actinomyces’ signatures were also observed [30,31]. On top of that, Costantini et al. also described Proteobacteria, Firmicutes, Actinobacteria and Bacteroidetes associated with breast tumors being the most prominent *Ralstonia* genus [32].

Importantly, Urbaniak et al. described a substantially different breast microbiota pattern comparing tumor and healthy breast tissue from women affected with BC to breast tissue from healthy controls. Patients with BC presented higher relative abundances of *Bacillus*, Comamondaceae, Bacteroidestes, Enterobacteriaceae and *Staphylococcus* compared to healthy controls. These last two groups of bacteria are capable of inducing DNA damage, possibly breaking the DNA double strand, while *Bacillus* has other carcinogenic effects including hormone metabolization and/or stimulation of cell proliferation. On the other hand, lower levels of some bacteria such as *Lactococcus* and *Streptococcus*, with anti-cancer properties, were found in women with BC compared to healthy controls [33]. Likewise, Hieken et al. found notable differences in β-diversity (variation of microbial communities between samples) when comparing the breast tissue microbiome from women with benign breast disease versus women with invasive BC. The breast tissue of women with cancer was significantly enriched in gender-specific taxa including *Fusobacterium, Atopobium, Gluconacterobacter, Hydrogenophaga* and *Lactobacillus* [34]. In addition, studying the functional role of these bacteria within microenvironments, six differentially abundant pathways were identified comparing patients with benign and malignant breast disease. In patients with BC, a significant over-expression of genes involved in the metabolism of cysteine and methionine, glycosyltransferases and fatty acid biosynthesis were observed [34]. Interestingly, methionine dependence is a general metabolic disorder in multiple cancers, and it is postulated that methionine reduction could reverse cancer progression [47].

### 3.2. Link between Gut Microbiota and Breast Cancer

There are also several studies in gut microbiota involving patients with BC (Table 1). For example, Bard et al. [37] observed significant differences in fecal microbiota, specifically in the absolute numbers of *Bifidobacterium* and *Blautia*, and the proportion of *Faecalibacterium prausnitzii* and *Blautia*, in relation to clinical stages of BC. They observed that stage I breast tumors had a lower absolute number of *Blautia* spp. than stage III. Furthermore, they found differences concerning the absolute number of bacteria according to BMI [37]. Luu et al. also determined that the gut microbiota composition of BC women was variable according to clinical stage and overweight/normal BMI. They found that overweight and obese women had a decrease in the total number of Firmicutes, *Faecalibacterium prausnitzii* and *Blautia* spp. comparing patients of normal weight. In addition, patients with stage II/III tumors had an increase in the total number of Bacteroidetes, *Clostridium coccoides* cluster, *Clostridium leptum* cluster, *Faecalibacterium prausnitzii*, and *Blautia* spp. than patients in stage 0/I [38]. On the other hand, Fruge et al. showed differences in gut microbiota related to elevated body fat, highlighting the prevalence of *Akkermansia muciniphila* in stage 0–II breast tumors. Additionally, in BC women with high relative abundance of *A. muciniphila*, higher abundance of *Prevotella* and *Lactobacillus* and lower of *Clostridium*, *Campylobacter* and *Helicobacter* were detected when compared to patients with low relative abundance of the bacteria [39].

Additionally, Goedert et al. described that postmenopausal women with BC had altered fecal microbiota and lower α-diversity (variation of microbes in a single sample) and this was independently associated with estrogen concentration. They observed that patients with BC had elevated levels of Clostridiaceae, *Faecalibacterium*, and Ruminococcaceae, and a decrease in the levels of *Dorea* and Lachnospiraceae than paired controls [40]. According to Zhu et al., postmenopausal BC women have their fecal microbiota enriched in *Escherichia coli*, *Citrobacter koseri*, *Acinetobacter radioresistens*, *Enterococcus gallinarum*, *Shewanella putrefaciens*, *Erwinia amylovora*, *Actinomyces* spp. HPA0247, *Salmonella enterica* and *Fusobacterium nucleatum*. However, they did not find differences between cases and controls in premenopausal women [41].

Nevertheless, the results obtained from these different studies do not allow us to make definitive conclusions considering the variations in study population in terms of age, ethnicity, geographical location, sequencing techniques as well as analysis methodologies.

## 4. Role of Gut Microbiota in Estrogen Level Regulation

It is known that sex hormone dysregulation of is one of the main risk factors for BC development. Hormonal deregulation manifests itself, both clinically and molecularly, in the different BC subtypes [48,49]. It has been demonstrated that a subset of microbes within the gastrointestinal tract influences estrogen metabolism and the balance of circulating and excreted hormone levels [50]. These microbes are collectively referred as estrobolome, capable of producing beta-glucuronidase enzymes altering estrogens into their active forms and increasing the availability of intestinal estrogens for resorption in the bloodstream [42]. Estrogens’ metabolism occurs in the liver, where they are conjugated and excreted into the gastrointestinal lumen through the bile. Then, bacterial β-glucuronidase de-conjugates them, and finally, they are re-absorbed as free estrogens through enterohepatic circulation. Using this pathway, free estrogens are distributed to different distant organs such as the breast.

There are several β-glucuronidase bacteria in Clostridia (*Clostridium leptum* and *Clostridium coccoides*), Ruminococcaceae families [15,51] and the *Escherichia*/*Shigella* bacterial group [15,52]. Recently, it has been demonstrated that postmenopausal estrogenic metabolism is associated with microbial fecal diversity [35]. In fact, the relative abundance in the order Clostridiales was correlated with the ratio of estrogen metabolites to parent estrogens, whereas the genus *Bacteroides* was inversely correlated [42]. Likewise, it has been proposed that estrogen conjugation by β-glucuronidase could be associated with the microbiota dysbiosis described in women with BC. In this respect, the beta-glucuronidase enzyme levels were higher in nipple aspirate fluid (NAF) from BC patients compared with healthy women. In this study, *Alistipes* was the bacteria genus most relatively abundant in NAF from women with BC, in contrast, an unclassified genus of the Sphingomonadaceae family was observed in healthy women [35].

Finally, studies using fecal samples from patients with BC demonstrated a positive correlation between the abundance of *Streptococcus* and the presence of β-glucuronidase and/or β-glucosidase enzymes, which cleave the estrogen glucuronide conjugate and promote recirculation of estrogen [53]. In addition, other estrogen-like metabolites can also be produced by oxidative and reductive reactions in the gut and by an induced synthesis of estrogen-inducible growth factors, which might have carcinogenic potential. Moreover, bacterial β-glucuronidase could participate in the deconjugation of xenobiotics and/or xenoestrogens, leading to their re-uptake through the enterohepatic pathway, thus increasing their half-life and availability [15,54].

## 5. Breast Cancer, Microbiota and the Immune System

BC was initially considered a non-immunogenic tumor. However, recent studies have shown that the expression of genes related to the immune system and the presence of immune infiltrates in primary tumors were associated with better clinical outcome [55]. This observation was particularly interesting since it involved the HER2+ and TNBC tumors, the most aggressive subtypes. In this regard, CD8+ T cells, which generally represent cytotoxic T cells, can directly kill cancer cells and their presence is associated with better prognosis [56]. In contrast, FOXP3+ CD4+ regulatory T cells act primarily by mediating immune tolerance and their presence correlates with a poor prognosis [57]. In BC, the percentage of Treg cells increases in parallel with the stage of the disease, from normal to in situ ductal carcinoma (DCIS) and from DCIS to invasive carcinoma [58]. In patients presenting invasive carcinomas, the presence of a high FOXP3+T cell number predicts for a shorter relapse-free survival and overall survival [59]. This is potentially indicating the presence of Treg cells with immunosuppressive characteristics which promote immune evasion and cancer progression.

The mechanisms by which growing tumors can stimulate Treg lymphocyte proliferation and differentiation are not well known, but the production of prostaglandin E2 by tumor cells and the cytokine CCL22 by tumor-associated macrophages can act as chemotactic and differentiation agents for these cells [60,61]. It has been suggested that the increase or decrease in the abundance of some specific bacteria in gut microbiota may result in a higher production of Tregs or reduce differentiation of pathogenic T cells, probably preventing inflammatory diseases [62]. For example, Treg cells expressing the FOXP3 transcription factor play an essential role in regulating the immune response of the commensal microbiota and the metabolites produced by these bacteria could regulate Treg cell turnover [63]. Furthermore, some bacterial metabolites, such as butyrate and propionate, have been shown to exert a potent anti-inflammatory effect through the modulation of colonic regulatory T cells in animal models [62,64,65]. In another study, Goedert et al. described a significant estrogen-independent association between IgA+ and IgA− gut microbiota in BC patients. When they compared BC patients with IgA+ and IgA− microbiota, those ones presenting IgA+ had significantly lower richness and α-diversity of their fecal microbiota than cases with IgA− microbiota. The estrogen-independent associations in IgA+ and IgA− gut microbiota are significantly different when comparing controls and postmenopausal BC women, suggesting that gut microbiota may influence BC risk by altering metabolism, estrogen recycling and immune pathways [15,43].

Taken together, these data indicate that microbial DNA present in the breast and the bacteria-derived metabolites could influence the local immune microenvironment. This means that our commensal bacteria could directly influence tumor processes using their metabolic capacity affecting immune cells and the inflammation process.

## 6. Breast Cancer, Microbiota and Inflammation

The mucosal surface barriers allow host/microbe symbiosis and due to its susceptibility to constant environmental aggressions, must be quickly repaired to restore homeostasis. Once these barriers are damaged, the microbes can influence immune responses to tumors causing proinflammatory or immunosuppressive microenvironments. In this respect, the mechanism by which gut bacteria can promote BC is through chronic inflammation, which is associated with tumor development [66]. Gut bacteria can upregulate the Toll-like receptors (TLR), and activate NF-kB, which is important in inflammation regulation and associates with cancer. In fact, the activation of NF-kB leads to the release of IL-6, IL-12, IL-17 and IL-18 as well as the tumor necrosis factor alpha (TNF-alpha), triggering persistent inflammation in the tumor microenvironment [67,68,69]. Likewise, the molecular Patterns Associated with Pathogenic Microorganisms (PAMP) are recognized by innate-immune system cells through Pattern Recognition Receptors (PRR) including the Toll (TLR) and Nod (NLR) receptors. These PAMPs are essential components for pathogens that permit their survival and contribute to their pathogenicity such as the bacterial lipopolysaccharides (LPS), flagellin, lipoteic acid, peptidoglycans and unmethylated CpG oligodeoxynucleotides. TLRs, by recognizing PAMPs, are capable of activating proinflammatory cytokines production from innate response cells. In fact, chronic activation of TLRs promotes tumor cell proliferation and improves invasion and metastasis mechanisms through the regulation of cytokines, metalloproteinases and proinflammatory integrins [70] (Figure 1). It has been described that TLRs play an important role in the initiation and promotion of BC. Regarding this, it has been shown that TLR5 receptors are highly expressed in breast carcinomas and its activation by the flagelin ligand leads to a potent anti-tumor activity and inhibits BC cell proliferation [71].

However, the pathogen-induced inflammation is not limited to the site of infection. It has been shown that C57BL/6 ApcMin/+ mice which are genetically predisposed to develop breast carcinomas do not develop breast tumors when they grow up in specific pathogen-free conditions [72]. However, after gastric administration of *Helicobacter hepaticus*, they developed mammary carcinomas as a result of the innate immune induction through inflammation [73,74]. Therefore, we can affirm that chronic inflammation influences the initiation as well as the progression of BC by the persistent presence of inflammatory cytokines and immune-cell recruitment, such as Tregs, which as above-mentioned, decrease immune response promoting tumor immune escape.

## 7. Breast Cancer, Microbiota and Epigenetic Regulation

In patients with BC, tumor-suppressor gene expressions are often inactivated through changes in epigenetic marks in response to environmental stimuli [75]. Epigenetic reprogramming has been implicated in the different subtypes of BC. For example, methylation in the promoter of the ERα gene has been seen in TNBC, associated with poor prognosis in women with no family history of BC. Another gene affected by epigenetic regulation is the BRCA1 gene, which predisposes women to ovarian or BC. The gut microbiome is a contributor frequently overlooked in epigenetic deregulation, which is capable of interacting physiologically and environmentally with the tumor as mentioned above. These microorganisms are able to produce low molecular weight bioactive substances such as folates, short chain fatty acids (butyrate and acetate) and biotin, which can participate in epigenetic processes [76]. For example, it has been shown the ability of butyrate to activate epigenetically silenced genes in cancer cells such as p21 and BAK [77]. On top of that, the intestinal microbiota also contributes to minerals’ absorption and excretion including zinc, iodine, selenium, cobalt and others, which are cofactors of enzymes participating in epigenetic processes. On the other hand, several enzymes such as methyltransferases, acetyltransferases, deacetylases, Bir A ligase, phosphotransferases, kinases and synthetases are derived from the intestinal microbiota [78].

However, although gut bacterial influence can promote hypermethylation and epigenetic reprogramming in humans and contribute to tumor processes, a direct cause of bacterial epigenetic activation capable of inducing breast tumor formation has not yet been proven. Therefore, fluctuations in gut microbiome composition, microbiome interaction with host´s immune system and subsequent chemical changes influenced by bacterial metabolic by-products, are potentially involved in epigenetic regulation.

## 8. Diet, Microbiota and Breast Cancer

Approximately 35% of all cancers are associated with dietary intake including 50% of breast carcinomas [79]. In diet-associated BC, microbial-mediated mechanisms are possible modulators of carcinogenesis and tumor aggressiveness. It has been described that a different factor shaping the gut microbiome or microbial diversity, but one of the major components, is the diet content and quality [80]. It has been described as the partial effect of different diets in the proliferation of the eubiotic microbiome having synergistic effects during cancer therapy [81]. Several studies have investigated the relationship between BC, microbiota and well-described diets such as the Mediterranean one. The Mediterranean diet is one of the most studied diets linking microbiota and BC. An increased intake of mono- and polyunsaturated fatty acids, gut microbiota-accessible carbohydrates and fruits, vegetables and legumes are related with an overall improvement in the health status that can be potentially protective reducing cancer risk and cancer mortality, also in BC [82,83].

Shively et al. demonstrated, using a non-human primate model (*Macaca fascicularis* monkey), that a Mediterranean diet compared to a Western diet had marked effects in mammary gland microbiota populations and metabolite profiles. The Mediterranean diet was associated with 10-fold higher breast tissue *Lactobacillus* abundance compared with mammary tissue from Western diet-fed monkeys (with high dietary intake of saturated fats and sucrose and low intake of fiber). Moreover, mammary glands from the Mediterranean diet group presented higher levels of certain bile acid metabolites and increased bacterial-processed bioactive compounds. In matched monkey groups, the analysis of plasma bile-acid metabolites showed no significant regulation of taurocholate and glycocholate or chenodeoxycholate by diet, suggesting a possible mammary-gland-specific microbial regulation of bile acid metabolites. Furthermore, the significant increase in the abundance of *Lactobacillus* in the mammary glands of Mediterranean diet-fed monkeys may increase mammary gland-specific bile acid metabolite-mediated activation of the farnesoid X receptor signaling potentially increasing anticancer properties [84].

In a recent study using overweight BC survivors, Pellegrini et al. assessed the efficacy of probiotics together with Mediterranean diet versus diet alone on gut microbiota and the metabolic profiles. They found that probiotics (*Bifidobacterium longum* BB536 and *Lactobacillus rhamnosus* HN001) in addition to Mediterranean diet significantly increase the bacterial diversity and decrease the Bacteroides/Firmicutes ratio as well as improved metabolic (fasting glucose and fasting insulin) and anthropometric parameters (body mass index, waist circumference and waist to hip ratio) compared with the Mediterranean diet alone [85]. All these findings could open a new avenue for BC prevention and treatment.

## 9. Probiotics Effects against Breast Cancer

Probiotics are live bacteria that can maintain healthy microbiota and restore a beneficial microbial composition [86]. One of the most significant characteristics of probiotics is the production of substances such as antibiotics, anticarcinogens, or other compounds with beneficial effects in general health and pharmaceutical properties [87] (Figure 2).

Several in vitro and in vivo studies investigated the effects of probiotics on BC. de Moreno de le Blanc et al. described the immunoregulatory capacity of milk fermented by *Lactobacillus helveticus* R389 on the immune response in mammary glands in the presence of local breast tumors. Mice fed with *L. helveticus* R389-fermented milk and injected with BC tumor cells, showed an increased in IL-10 and a decrease in IL-6 cytokine levels in serum and mammary cells of mice, leading also to breast tumor cell inhibition [88]. In this regard, Yazdi et al. studied the effects *Lactobacillus acidophilus* oral administration on the immune responses using BALB/c mice transplanted with a breast tumor. The authors suggest that daily consumption of *L. acidophilus* can increase the production of immunomodulatory cytokine IL-12 in the splenocyte culture, while the tumor growth rate in the mice decreased [89]. Lakritz et al. showed that the oral intake of the probiotic *Lactobacillus reuteria* inhibited early stages of BC in two mice models, one group with genetic predisposition to BC and the other group fed with a Western-style diet to develop mammary tumors [90]. In addition, Kassayová et al. observed that long-term administration of *Lactobacillus plantarum* LS/07 is effective against BC through immunomodulatory mechanisms [91].

Additionally, Imani Fooladi et al. showed that daily oral administration of *L. acidophilus* two weeks before BC tumor transplantation and continuation for 30 days, produced a significant increase in the overall survival, suggesting that *L. acidophilus* could promote immune responses and may increase the antitumor response [92]. Zubaida et al. investigated the potential of heat-killed cells (HKC), the cytoplasmic fractions (CF) of *Enterococcus faecalis* and *Staphylococcus hominis* as anti-breast cancer agents in the MCF-7 cell line. The two forms of the bacteria caused a significant decrease in MCF-7 cell proliferation, induction of apoptosis and cell cycle arrest in a concentration- and time-dependent manner [93]. In another study, Zamberi et al. analyzed the antimetastatic and antiangiogenic effects of kefir water made from kefir grains in mice inoculated with 4T1 BC cells. They found that Kefir water inhibited tumor proliferation, promoting cancer cell apoptosis, modulated the immune system, and had anti-inflammatory, antimetastatic, and antiangiogenesis effects [94].

In clinical trials, probiotics showed benefits in quality of life, therapy-related toxicities and post-operative complications in colon cancer patients [95] and other cancer types [96]. However, only a few clinical trials have been developed using BC patients. In Japanese women, regular consumption of *L. casei* Shirota and soy isoflavone from adolescence was significantly associated with decreased BC risk, showing chemopreventive effects on cancer development. Nevertheless, this study needs long-term exposures and surveillance to correlate cancer chemopreventive effects to these bacteria consumption [97]. Only two clinical trials related to the benefit of probiotics in BC patients are registered in the clinicaltrial.gov web page. In the study NCT03358511, twenty post-menopausal BC patients took the probiotic Primal Defense^®^ ULTRA (Garden of Life LLC, Jacksonville, FL, USA) for 2–4 weeks/ three times a day prior to surgery in operable stage I–III breast adenocarcinoma tumors. The study is ongoing and the role of probiotics of cytotoxic T lymphocytes (CD8+ T-cells) numbers in BC patients will be investigated. A second study (NCT03760653), is a randomized controlled pilot study to determine the effects of probiotics supplementation (*Lactobacillus rhamnosus*, *Lactobacillus paracasei*, *L. acidophilus*, and *Bifidobacterium bifidum*) administered over 12 weeks together with physical exercise on the bacterial balance from the gut, the gastrointestinal immune system as well as the quality of life in BC survivors.

Although preclinical studies showed that probiotics could be proposed moderators to prevent and control BC progression enhancing the host’s immune system, additional efforts through clinical trials or prospective studies are necessary to establish the probiotics’ efficacy in the clinical management of BC patients, uncovering also the immune system mechanisms.

## 10. Prebiotics, Microbiota and Breast Cancer

Prebiotics are substances that enhance the growth or activity of gut microorganisms and are typically non-digestible dietary fiber compounds [98]. These components of dietary fiber, combined with harmful and carcinogenic substances in the gut, promote their discharge and decomposition [99] and increase the growth of probiotics inhibiting the proliferation of pathogenic bacteria and production of carcinogens [100] (Figure 2). Dietary fiber may alter the gut microbiota and influence estradiol metabolism through specific enzyme activities, such as β-glucuronidase in postmenopausal BC patients [101]. Zengul et al. examined the relationship between dietary fiber and gut microbiota in their role increasing the β-glucuronidase activity and circulating estrogens in post-menopausal BC patients. The results of this study indicated that dietary fiber intake had no correlation with the estrogen levels in the blood. However, they found that higher levels of total and soluble dietary fibers correlated with lower levels of *Clostridium hathewayi* spp. and *Clostridium* (Erysipelotrichaceae family), which promote β-glucuronidase activity [102].

Plant-based lignans are present in high concentrations in soy, flax seeds and sesame, and to some extent in fruits, vegetables and berries [103]. These lignans are converted into compounds such as enterolactone, the most prevalent phytoestrogen produced by the *Eggerthella* action [104]. Phytoestrogens which are compounds structurally and functionally similar to mammalian estrogens, exert their effects on BC by inhibiting estrogen synthesis and metabolism, as well as, through their antiangiogenic, antimetastatic and epigenetic effects. Moreover, the effect of phytoestrogens was comprehensively evaluated with respect to BC recurrence and survival [105]. They can reduce the levels of estrogen in the blood by inhibiting the aromatase enzymatic activity [106]. In a recent meta-analysis, an important inverse association between serum enterolactone and postmenopausal BC risk, which was more evident in ER−PR− than ER+PR+ tumors, independently of the HER2 status, has been observed [107]. Similar results were described in gnotobiotic rats colonized by lignan-converting bacteria *Clostridium saccharogumia*, *Eggerthella lenta*, *Blautia producta* and *Lactonifactor longoviformis*, where the synthesis of enterolignans from dietary lignans by gut microbes had protective effects against BC development [108]. Fink et al. studied the relationship between the pre-diagnosis phytoestrogen (isoflavones) intake and breast cancer survival in a cohort of US pre- and post-menopausal BC patients. The authors showed a lower risk of BC mortality in the highest versus the lowest quintile of isoflavone intake but only in postmenopausal women [109]. On the other hand, Boyapati et al., conducted a study during a 5-year period prior to diagnosis in pre- and post-menopausal Chinese BC patients. In this study, no overall association between soy intake prior to cancer diagnosis and disease-free BC survival for women in the highest tertile of intake compared with those in the lowest tertile was found. This association was not influenced by the ER/PR status, TNM staging, age at diagnosis, body mass index, waist to hip ratio and evaluation for genetic polymorphisms for ERa and ERb [110,111]. Similarly, in the DietCompLyf study, it was evaluated the associations between phytoestrogen intake levels and BC recurrence and survival in UK women diagnosed with grades I–III BC over 5 years. No significant associations between pre-diagnosis phytoestrogen intake and improved BC prognosis were observed [112]. Finally, Verheus et al. investigated the association between plasma phytoestrogen levels and breast cancer risk in a cohort of Dutch pre- or perimenopausal and postmenopausal women who developed BC. They showed that high genistein circulation levels are associated with reduced BC risk [113].

SCFAs such as acetate, propionate and butyrate are produced by bacterial fermentation of dietary fiber in the colonic lumen. The anti-cancer effect of butyrate was demonstrated in cancer cell cultures and animal models [114]. A study from Salimi et al. revealed, using the MCF-7 and MDA-MB-468 cell lines, that sodium butyrate deceased the rate of viable cells in a dose and time dependent manner, which was correlated with cell cycle arrest and induction of apoptosis accompanied by an elevated level of ROS and mitochondrial disruption [115]. In another study, Kim et al. investigated the effects of combined therapy of 5-aza-2′-deoxycytidine (5-aza-CdR), sodium butyrate and tamoxifen in tumor suppression and expression of anti- or proapoptic proteins in BC cell lines (MDA-MB-231 and MCF-7). The study showed that the combined therapy with 5-aza-CdR, sodium butyrate and tamoxifen was the most effective to produce apoptosis in BC cells, changing the expression of pro-apoptotic regulator proteins [116]. In addition, Wang et al. investigated the role of sodium butyrate on MCF-7 cells and indicated that sodium butyrate, at certain concentrations, induced MCF-7 cell apoptosis. Moreover, they characterized the morphology of these cells presenting thick nucleoli, chromatin margination, reduced mitochondria and dramatic vacuoles [117].

Dietary polyphenols are also natural compounds produced by plants, and present in fruits, vegetables, cereals, tea, coffee and wine. Polyphenols are biotransformed by the gut microbiota into derivates resulting in an increased bioavailability. On the other hand, polyphenols are capable of modulating the composition of the gut microbial community mostly by inhibiting the proliferation of pathogenic bacteria and stimulating the beneficial counterparts [118]. In this regard, also Sharma et al. described the effects of broccoli sprouts (BSp) and green tea polyphenols (GTPs) consumption, early in life, alone or in combination in the gut microbiota and SCFAs metabolism in a Her2/neu transgenic mice model of BC known to spontaneously develop ER-mammary tumors. The authors showed that the group consuming both elements presented the strongest inhibiting effect on tumor volume and a significant increase in tumor latency. Moreover, at taxa level, more presence of the genera *Allobaculum*, *Lactococcus*, Ruminococcaceae and S24-7 family in BSp-fed and in combination-fed mice were observed when administered early in life [119]. In another study, it was analyzed the administration of epigallocatechin-3-gallate (EGCG) (a polyphenol present in tea) to inhibit cell proliferation, invasion and angiogenesis in BC patients undergoing treatment with radiotherapy. They found that EGCG increased the efficacy of radiotherapy in the patients [120]. Sheng et al. demonstrated that EGCG significantly decreased the SCUBE2 methylation status by reducing DNA methyltransferase expression and activity, resulting in the inhibition of BC progression [121].

Then, the use of dietary bioactive compounds as an adjuvant therapy for chemoprevention of BC could be of great interest and could open new avenues for preventive procedures in BC.

## 11. Antibiotics, Microbiota and Breast Cancer

Antibiotics are compounds used to eliminate all bacterial populations without discrimination between pathogenic or beneficial. They cause the so-called gut dysbiosis, which includes loss of taxonomic richness or reduction in commensal community members, that favor the expansion of opportunistic pathogens, which will also affect microbiota functioning and host-microbial inter-dependence inducing detrimental health effects [122]. Treatments using antibiotics in cancer patients are a hot topic of discussion since they are frequently prescribed alongside chemotherapy and cancer surgery. It has been observed in some types of cancer, such as BC and melanoma, that antibiotics can accelerate the progression of the disease [123]. However, in other tumor types such as pancreatic cancer they had positive effects [124]. In addition, some other studies showed a positive correlation between the use of antibiotics in days and the risk of BC when adjusted for age and length of enrollment [125]. In a very recent meta-analysis, the authors observed that the type of antibiotic might be associated with BC risk. In this study, the risk was modestly increased when the patients were treated with penicillin, tetracycline and nitrofuran and marginally increased when nitroimidazole and metronidazole were used [126].

Finally, antibiotics have also been associated with gut microbiota disruption by decreasing response to platinum-based chemotherapies as well as immunotherapies [127,128] suggesting that an intact microbiome is necessary for optimum responses to anti-cancer therapies [127]. Regarding this, a study using BC mice models showed a reduction of butyrate in feces after antibiotic administration together with an increase in tumor growth. The observed butyrate reduction was produced by a decrease in the abundance of *Odoribacter* and *Anaeotruncus* genera, which are butyrate producing bacteria, as well as an increase in the abundance of *Bacteroides* [46]. Overall, it was shown that patients with enhance diversity of fecal microbiome experience significantly longer progression-free survival when compared to those with low or moderate microbiota diversity [127,129]. Therefore, designing antibiotics targeting a particular spectrum in the microbiome might help to regulate the gastrointestinal microbiome reducing BC risk.

## 12. Microbiota, Anti-Cancer Therapy and Disease Progression

The gut microbiota play an important role in the metabolism of chemotherapeutic drugs, activating or inactivating them. The presence of one type of gut microbiota or another can determine the degree of efficacy of a drug [130] and potentially exerting a substantial influence in the clinical guidelines to treat BC patients. Nevertheless, therapy can also modulate the breast microbiome as shown by Chiba et al. in women treated with neoadjuvant chemotherapy. They found a significant increase in the abundance of *Pseudomonas* spp. in breast tumor tissue as well as a bacterial diversity reduction. In this study, they also showed that non-treated patients had a lower abundance of *Prevotella* in tumor tissue [36]. On top of that, a study employing an HR+ BC mice model identified commensal dysbiosis as a host-intrinsic factor associated with metastatic dissemination. They observed a disruption in the commensal gut microbiota homeostasis resulting in an increased of circulating tumor cells dissemination as well as enhanced early inflammation in the mammary gland [44]. On the other hand, Horigome et al. described an association of polyunsaturated fatty acids (PUFAs) with some gut bacteria taxa like phylum Actinobacteria and Bacteroidetes in patients previously treated with chemotherapy and an association of the genus *Bifidobacterium* with non-treated participants [45].

An influence of the intestinal microbiota on the efficacy or toxicity of some chemotherapeutic drugs such as cyclophosphamide, platinum salts and irinotecan has been determined [131]. In this regard, cyclophosphamides can damage the gut mucosa, making the gut permeable for gut bacteria, permitting their access into the bloodstream [132]. On the other hand, the association of the probiotic bacteria *Lactobacillus plantarum* HY7712 in gut microbiota with a protective role against cyclophosphamide-induced immunosuppression using mice models has been described [133]. Anthracyclines are also metabolized by several gut bacteria such as *Streptomices* WAC04685, which are capable of inactivating doxorubicin by deglycosylation [134]. Furthermore, upon anthracycline treatment, commonly Gram-positive microorganisms such as *Lactobacillus johnsonii*, *Lactobacillus murinus*, *Barnesiella intestinihominis* and *Enterococcus hirae* can pass the intestinal barrier to enter secondary lymphoid organs, thereby influencing the anticancer immune response from the host [135,136,137]. Overall, it has been well demonstrated that the gut microbiome can interfere with anthracyclines’ bioavailability modifying their pharmacokinetics and pharmacodynamics [5].

Likewise, taxanes are subjected to bacterial metabolism [138] and selective estrogen receptor modulators (SERMs) such as Tamoxifen and Raloxifen can change the composition of the microbiome [139]. SERMs can be toxic for *Pseudomonas aeruginosa*, *Klebsiella pneumoniae*, *Acinetobacter baumannii* [140], *Porphyromonas gingivalis*, *Streptococcus mutans* [141], *Enterococcus faecium* [142], and *Bacillus stearothermophilus* [143]. Additionally, taxanes can also interfere with bacterial LPS activating the immune system [144].

Recently, it has also been shown that the composition of the intestinal microbiota has an important influence in anticancer immune effectiveness and in action mechanisms of immunogenic chemotherapies. In fact, some bacteria such as *A. muciniphila*, *Bacteroides fragilis*, *Bifidobacterium* spp. and *Faecalibacterium* spp., have been associated with favorable anticancer immune responses in both animal models and cancer patients. Importantly, these bacteria also appear to have a positive influence in general health, reducing the incidence of metabolic disorders and a wide range of chronic inflammatory pathologies [145]. The intestinal microbiota also seem to play a key role in the development and severity of mucositis, one of the most common side effects of the gastrointestinal system in patients undergoing chemotherapy. The intestinal microbiota may attenuate or aggravate mucositis by influencing inflammatory processes [146].

## 13. Conclusions

Numerous studies have linked the microbiome with the initiation and progression of different tumor types. Although traditionally some tissues such as breast have been considered as “sterile”, it has been recently observed that they contain a varied bacterial population. In this regard, an alteration of the human microbiota population is called dysbiosis and characterizes BC. Several studies have shown that the microbiota present in breast tumor tissue differs from those in adjacent normal tissue but also from the tissue of healthy individuals. In this regard, the variation in the abundance of specific pathogens in the different BC subtypes and clinical stages has been also identified, such as the increase of *Methylobacterium radiotolerans* in ER+ BC patients or the decrease in *Blautia* spp. in stage I BC patients compared to stage III, but the contribution mechanism to BC development is still unknown. Gut microbiota can interfere in estrogen metabolism through bacteria since they present β-glucuronidase enzymes and can de-conjugate estrogens to free estrogens, which through enterohepatic circulation, are delivered at distant parts of the body such as the breasts.

On the other hand, the microbiota play a fundamental role in the development of the host´s immune system and tumor immunity. In addition, the metabolites produced by commensal bacteria such as butyrate and propionate (SCFAs) can control the development of colonic regulatory T (Treg) cells by upregulating the Toll-like receptors (TLR), activating the nuclear factor kB (NF-kB), releasing interleukins including IL-6, IL-17 and TNF-alpha and finally leading to a persistent inflammation in the tumor microenvironment. Regarding this, it has been shown that chronic inflammation influences the initiation as well as the progression of BC by the constant presence of pro-inflammatory cytokines and immune cell recruitment such as Tregs, which on top of that, decrease the immune response. These SCFAs could also promote hypermethylation and epigenetic reprogramming in BC, although this affirmation has not been proved yet. Antibiotic administration has been associated with gut microbiota disruption and increased risk of incident BC, but the effect depends on the type of antibiotics. Nevertheless, several in vivo and in vitro studies show remarkable evidence that diet, probiotics and prebiotics could exert important anticarcinogenic effects in BC and could also be used as adjuvants to conventional treatment for BC. Additionally, gut microbiota are also involved in chronic inflammation when the mucosal barrier is weakened or broken due to anti-cancer treatment, contributing to the aggravation of mucositis.

However, there is still much to determine in relation to gut microbiota action mechanisms as well as the investigation of suitable strategies to modulate and sustain changes in gut microbiota, for example via dietary and prebiotic or probiotic supplementation to improve responses to cancer therapy. In addition, immunomodulator drugs with probed efficacy in cancer treatment are prone to modify patients’ microbiome, and their characterization could offer important insights in the different observed disease responses to treatments. It is also necessary to understand, using mouse models and BC patients, how different antibiotic regimens can induce disturbances in breast/gut microbiota and their influence in BC progression.

In summary, it is necessary not only to study the association between gut microbiota and anti-tumor immune responses using metagenomic sequencing technologies, but also to demonstrate microbiota mechanisms of action applying transcriptional and/or metabolic profiling. Considering all the above, gut-tumor axis microbiota will definitely become important players in paving the way for precision medicine to treat BC patients.

## Figures and Tables

**Figure 1 cancers-12-02465-f001:**
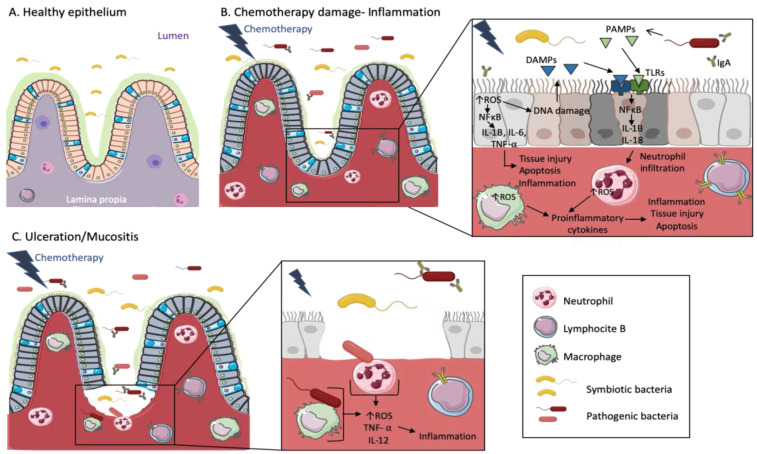
Chemotherapy treatments cause inflammation and mucositis in the intestinal epithelium of cancer patients. The cartoon representing a healthy epithelium (**A**) shows a wide diversity of symbiotic bacteria in the lumen and some inactive immune cells in the lamina propia. Both spaces are separated by a mucosal barrier. When patients are treated with chemotherapy (**B**), the mucosal barrier is damaged and pathogenic bacteria coexist with symbiotic bacteria in gut microbiota. Then, intestinal epithelial cells suffer DNA damage and tissue injury/cell apoptosis mediated by the increase of Reactive Oxygen Species (ROS) and cytokines signaling cascades. Cells affected by DNA damage release Damage-Associated Molecular Patterns (DAMPs), which together with Pathogen-Associated Molecular Patterns (PAMPs) released by pathogenic bacteria, are recognized by Toll-like receptors (TLRs). This signal is transmitted by the NFκB pathway, leading to the release of cytokines such as IL-1B, which enhance neutrophil/macrophage infiltration. ROS levels are increased in macrophages and neutrophils producing proinflammatory cytokines, tissue injury and apoptosis. On the other hand, B-lymphocytes produce IgA against the pathogenic bacteria. When the mucosal barrier is broken and the tissue injury advances (**C**), pathogenic bacteria interact with immune cells, increasing ROS production and provoking the activation of TNF- α as well as cytokines production, such as IL-12, also triggering inflammation.

**Figure 2 cancers-12-02465-f002:**
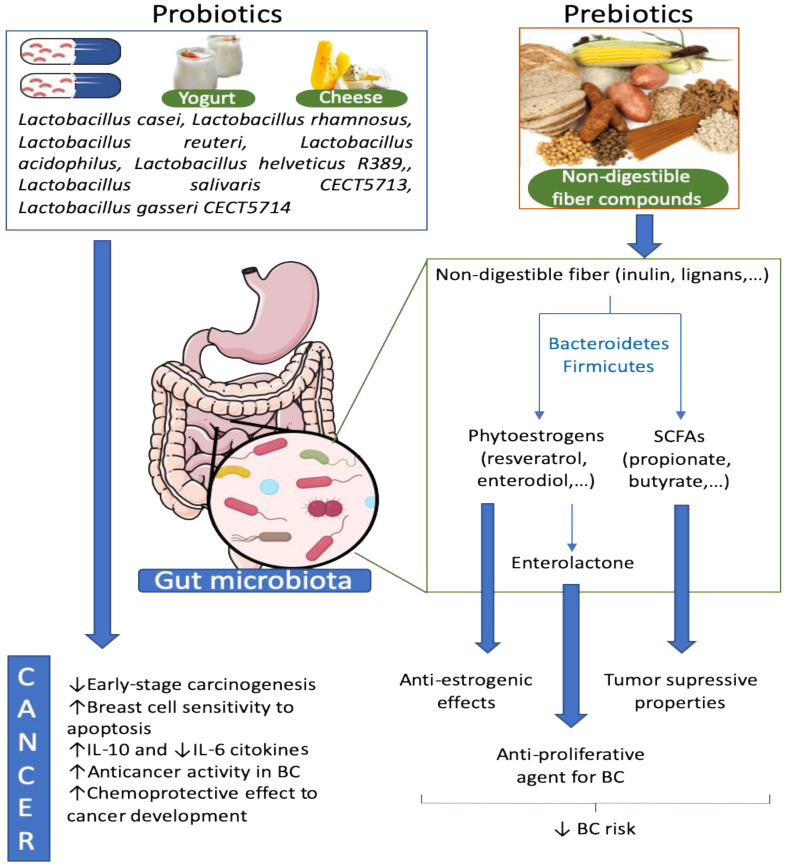
Effect of probiotics and prebiotics on the gut microbiota and BC. Probiotics are living beneficial bacteria, which can restore a dysbiotic microbiota. Some species of *Lactobacillus* were described to have anticancer activities in BC among other positives effects. On the other hand, prebiotics are normally non-digestible fibers that enhance the proliferation of beneficial bacteria in the gut. Non-digestible fibers can be converted to phytoestrogens and SCFAs by some bacteria belonging to Bacteroidetes and Firmicutes phyla. These bacteria-derived metabolites, as well as other derivates, have tumor suppressor properties and anti-estrogenic and anti-proliferative effects that reduce BC risk.

**Table 1 cancers-12-02465-t001:** Summary of studies addressing the association between breast cancer (BC) and breast-gut axis microbiota.

Study	Microbiome Tissue Related	Cohort	Sample Type	Main Methodology	Most Relevant Results
Xuan, C. et al., 2014 [25]	Breast	20 patients ER+ BC	Breast tumor tissue and its paired normal adjacent tissue	Pyrosequencing V4 16S rDNA Pipeline: QIIME	↑Proteobacteria, Firmicutes, Actinobacteria, Bacteroidetes and Verrucomicrobia (96.6%) in breast tissue. ↑*Methylobacterium radiotolerans* in BC tissue. ↑*Sphingomonas yanoikuyae* in paired normal tissue.
Urbaniak, C. et al., 2014 [22]	Breast	43 Canadian women (11 with benign tumors, 27 cancerous tumors and 5 healthy individuals) and 38 Irish women (33 women with BC and 5 healthy individuals)	Breast tissue coming from lumpectomies, mastectomies and breast reductions	V6 16S rRNA sequencing (Ion Torrent) Pipeline: UCLUST	↑Proteobacteria and Firmicutes in breast tissue. ↑*Bacillus* (11.4%) and *Acinetobacter* (10%) in Canadian women. ↑Enterobacteriaceae (30.8%) and *Staphylococcus* (12.7%) in Irish women. ↑*Escherichia coli* in BC tissue.
Yazdi, H.R. et al., 2016 [26]	Breast- Sentinel lymph	123 sentinel lymph nodes and 123 normal adjacent breast tissue samples	Sentinel lymph nodes and breast tissue	RT-PCR and pyrosequencing	↑*Methylobacterium Radiotolerance* in lymph cancer nodes samples compared to normal adjacent samples.
Wang, H. et al., 2017 [27]	Breast	57 women with invasive breast carcinoma and 21 healthy women	Urine Bilateral breast tissue from control patients underwent cosmetic procedures Tumor and ipsilateral adjacent normal breast tissue for cases by mastectomy	V3-V4 16S rRNA sequencing (Illumina) Pipeline: UCLUST	↓*Methylobacterium* and ↑*Corynebacterium*, *Staphylococcus*, *Actinomyces* and Propionibacteriaceae in patients with invasive breast carcinoma compared to healthy individuals.
Thompson, K.J. et al., 2017 [28]	Breast	668 tumor tissues (HER2+, ER+, TNC) and 72 normal adjacent tissues from The Cancer Genome Atlas (TCGA)	Breast tumor tissues and normal adjacent tissues	V3-V5 16S rRNA amplified sequencing data	↑Proteobacteria, Actinobacteria and Firmicutes in breast tissues. ↑Proteobacteri*a*, *Mycobacterium fortuitum* and *Mycobacterium phlei* in BC samples. ↑Actinobacteria in normal adjacent tissue.
Meng, S. et al., 2018 [29]	Breast	22 Chinese patients with benign tumor and 72 malignant BC patients	Breast tissue	V1-V2 16S rRNA sequencing (Illumina HiSeq)	↑*Propionicimonas*, Micrococcaceae, Caulobacteraceae, Rhodobacteraceae, Nocardioidaceae and Methylobacteriaceae, in BC tissues (ethno-specific) ↓Bacteroidaceae and ↑ *Agrococcus* are related with malignancy
Banerjee, S. et al., 2018 [30]	Breast	20 normal breast tissue and 148 BC tissue (50 ER or PR+, 34 HER2+, 24 TP and 40 TN)	Breast tissues	Pathochips array	↑Proteobacteria ↑*Actinomyces* in the four BC subtypes studied.
Banerjee, S. et al., 2015 [31]	Breast	100 women with triple negative BC (TNBC), 17 matched controls and 20 non-matched controls	Breast tissue. Matched controls were obtained from the adjacent non-cancerous breast tissue of the same patients with BC and non-matched were from different healthy women.	PathoChip array	↑*Brevundimonas diminuta, Arcanobacterium haemolyticum, Peptoniphilus indolicus, Prevotella nigrescens, Propiniobacterium jensenii and Capnocytophaga canimorsu*s in TNBC. Among virus, ↑ *Herpesviridae, Retroviridae, Parapoxviridae, Polyomaviridae, Papillomaviridae* in TNBC.
Costantini, L. et al., 2018 [32]	Breast	16 Mediterranean patients with BC (12 samples were collected from core needle biopsies (CNB) and 7 from surgical excision biopsies (SEB); 3 patients were processed with both procedure)	Fresh tumor breast tissue and paired breast healthy tissue	V3 16S-rRNA gene amplicons sequencing (Ion Torrent)	↑*Ralstonia* in breast tissue. No significant differences between healthy adjacent breast tissues and BC tissues.
Urbaniak, C. et al., 2016 [33]	Breast	58 women: 13 benign, 45 cancerous tumors and 23 healthy women	Breast tissue	V6 16S rRNA gene sequencing (Illumina MiSeq) Pipeline: QIIME	↑*Bacillus,* Enterobacteriaceae, *Staphylococcus,* Comamondaceae and *Bacteroidetes* and ↓ *Prevotella*, *Lactococcus, Streptococcus, Corynebacterium* and *Staphylococcus* in BC patients compared to healthy controls.
Hieken, T.J. et al., 2016 [34]	Breast	28 women undergoing non-mastectomy breast surgery: 13 benign breast disease and 15 invasive BC (100% ER/PR+ and 29% HER2+)	Breast tissue and breast skin	V3-V5 16S rDNA hypervariable taq sequencing (Illumina MiSeq) Pipeline: IM-TORNADO	↑*Fusobacterium*, *Atopobium, Gluconacetobacter*, *Hydrogenophaga* and *Lactobacillus* in BC tissue compared to healthy breast tissue.
Chan, A.A. et al., 2016 [35]	Breast	25 women with breast ductal cancer and 23 healthy women	Nipple aspirate fluid (NAF) and aerolar breast skin	V4 16S rRNA gene sequencing (Illumina MiSeq) Pipeline: Mothur	↑*Alistipes and* ↓ unclassified genus of the Sphingomonadaceae family in NAF from women with BC compared to healthy controls.
Chiba, A. et al., 2019 [36]	Breast	15 women with BC who were treated with neoadjuvant chemotherapy, 18 women with no prior therapy at time of surgery and 9 women who had tumor recurrence	Snap-frozen breast tumor tissue	V4 16S rRNA amplicon sequencing (Illumina Miseq) Pipelinee: Mothur (v.1.39.5) Microarray for confirmation	↑*Pseudomonas* spp. in BC tissue after neoadjuvant chemotherapy. ↓*Prevotella* in the tumor tissue from non-treated patients. ↑*Brevundimonas* and *Staphylococcus* in the primary breast tumors in patients developing distant metastases.
Bard, J.M. et al., 2015 [37]	Gut	32 BC women: Invasive ductal (81%), stage 0 (46,9%), grade II (62,5%), ER/PgR+ (80%), HER2+ (15%)	Fecal samples	PCR detecting 16S rRNA gene specific sequences	↓*Blautia* spp. in stage I compare to stage III BC. Significant differences in the abundance of *Bifidobacterium Blautia*, and *F. Prausnitzii* between clinical stages.
Luu, T.H. et al., 2017 [38]	Gut	31 women with early-stage BC (ER/PgR+ 90% and HER2+ 15%): 15 stage 0, 7 stage I, 7 stage II and 2 stage III. In total, 8 patients were overweight	Fecal samples	Real-time qPCR targeting specific 16S rRNA sequences	↑Firmicutes, *F. prausnitzii* and *Blautia* spp. in overweight and obese women compared to normal weight patients. ↑*Bacteroidetes, Clostridium coccoides* cluster, *Clostridium leptum* cluster, *Faecalibacterium prausnitzii*, and *Blautia* spp. in patients with stage II/III BC compared to patients in stage 0/I.
Fruge, A.D. et al., 2018 [39]	Gut	32 women with BC stage 0 to II	Fecal samples	V4 16S rRNA gene sequencing (Illumina Miseq)	↓*Akkermansia muciniphila* (AM) in BC patients with elevated body fat. ↑*Prevotella* and *Lactobacillus* and ↓ *Clostridium, Campylobacter* and *Helicobacter* in patients with high abundance of AM compared to patients with low abundance of AM.
Goedert, J.J. et al., 2015 [40]	Gut	48 postmenopausal women with BC and 48 paired control women	Urine and fecal samples	V3-V4 16S rRNA sequencing (Illumina) Pipeline: QIIME	↑Clostridiaceae, *Faecalibacterium*, and Ruminococcaceae and ↓ *Dorea* and Lachnospiraceae in BC patients compared to controls.
Zhu, J. et al., 2018 [41]	Gut	18 premenopausal BC patients, 25 premenopausal healthy control, 44 postmenopausal BC patients and 46 postmenopausal healthy controls	Fecal samples	Illumina sequencing	↑*Escherichia coli, Citrobacter koseri, Acinetobacter radioresisten*s, *Enterococcus gallinarum, Shewanella putrefaciens, Erwinia amylovora, Actinomyces* spp. *HPA0247*, *Salmonella enterica*, and *Fusobacterium nucleatum* and ↓*Eubacterium eligens* and *Roseburia inulinivorans* in postmenopausal BC patients.
Fuhrman, B.J. et al., 2014 [42]	Gut	60 healthy postmenopausal women	Urine and fecal samples	Pyrosequencing V1-V2 16S rRNA Amplicons Pipeline: QIIME	↑*Clostridiales* and ↓*Bacteroides* related with the ↑ ratio all estrogen metabolites to parent estrogens.
Goedert, J.J. et al., 2018 [43]	Gut	48 postmenopausal women with BC (11 women stage 0, 25 stage I, 10 stage II, 2 stage III; 88% ER+) and 48 paired control healthy women	Urine and fecal samples	V4 16S rRNA gene amplicon sequencing (Illumina MiSeq). Data available in the Sequence Read Archive under BioProject ID PRJNA383849	↓α- diversity and altered microbiota composition of both IgA-positive and IgA-negative fecal microbiota of BC patients.
Buchta Rosean, C. et al., 2019 [44]	Gut and mammary Breast (mice)	A mouse model of HR+ mammary cancer (5-to-8- week-old)	Feces and mammary tissue of mice	Flow cytometry Custom multiplex U-PLEX 16S rDNA sequencing (released by University of Maryland Institute for Genome Science)	↑Commensal dysbiosis. ↑Circulating tumor cells and metastatic dissemination. ↑Early inflammation within the mammary gland.
Horigome, A. et al., 2019 [45]	Gut	124 participants (46% history of chemotherapy) (123 women and 1 man)	Capillary blood and fecal samples	V3-V4 region of the bacterial 16S rRNA gene sequencing (Illumina Miseq) Pipeline: QIIME2 Gas chromatography for Fatty acid composition	↑Actinobacteria and Bacteroidetes are associated to polyunsaturated fatty acid (PUFAs) in patients previously treated with chemotherapy. ↑*Bifidobacterium* is associated to PUFAs in participants with no history of chemotherapy
Kirkup, B.M. et al., 2019 [46]	Gut (mice)	Female C57BL6 mice	Fecal samples	V1+V2 16S rRNA gene sequencing by Illumina (MiSeq) Nuclear Magnetic Resonance (NMR) spectroscopy	↓*Odoribacter* and *Anaeotruncus* (butyrate producing bacteria genera) and ↑ *Bacteroides.* ↓Butyrate in feces from BC patients treated with antibiotics and ↑ tumor growth.

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
