# Peer review of "Breast and Gut Microbiota Action Mechanisms in Breast Cancer Pathogenesis and Treatment"

_cancers, 2020, doi:10.3390/cancers12092465_

Round 1
Reviewer 1 Report
Well written and comprehensive review of an emerging and important topic.
I suggest adding a brief discussion early on of the approaches to assess microbiome (16S versus RNA seq etc.. ), and need for information on diet and colleciton details for samples and the pros and cons of techniques as well as which references use what technique.
When details specific to the field come up for the first time (ie alpha diversity) a brief explaination for a new reader.
Consider reformatting the table so comments/labels (they found.., major phyla) are in one column and another column is a list of phyla/taxa without the other words. It makes it easier to compare the dominant finding across studies.
Given the lack of clinical evidence for probiotics and prebiotics consider editing this language to say these are "proposed" moderators of... a little less definitive. It might be worthwhile to mention some of the non-breast cancer clinical studies in this space to add caution.
The data for the role of phytoestrogens and cancer risk versus cacner recurrence are variable and not high level. It would be worthwhile to be more inclusive of this uncertainty - ie https://pubmed.ncbi.nlm.nih.gov/18614351/
It would strengthen the conclusion to discuss top consistent findings from the review, which taxa come up more than once, any ongoing trials looking at immunotherapy effiacy based on microbiome in breast cancer, main gaps.
Some minor editing for english would strengthen it.
ie line 67, overweight or obesity should be corrected to being overweist or obese..
Line 72 one example... is followed by two examples.
220, "data" are plural
The legend for Figure 1 needs editing attention
Author Response
-Comment: Well written and comprehensive review of an emerging and important topic.
Response: We thank the reviewer for the comment.
-Comment: I suggest adding a brief discussion early on of the approaches to assess microbiome (16S versus RNA seq etc..), and need for information on diet and collection details for samples and the pros and cons of techniques as well as which references use what technique.
Response: Many thanks for bringing this important issue to our attention. As indicated by the reviewer, we have added a brief paragraph in the manuscript called “Approaches to assess microbiome: pros and cons”, page 2, lines 61-92, describing the main approaches to assess microbiome, the pros and cons of techniques and the need for information on diet and collection details for samples.
-Comment: When details specific to the field come up for the first time (ie. alpha diversity) a brief explanation for a new reader.
Response: Thanks for the suggestion. We have added a brief explanation to α- and β-diversity (Page 4, line 151 and line 180).
-Comment: Consider reformatting the table so comments/labels (they found … major phyla) are in one column and another column is a list of phyla/taxa without the other words. It makes it easier to compare the dominant finding across studies.
Response: Thanks for the interesting comment. As suggested by the reviewer, in order to make the table clearer and more informative, we have reformatted the table 1 and included a column to highlight the principal findings across studies.
Comment: Given the lack of clinical evidence for probiotics and prebiotics consider editing this language to say these are "proposed" moderators of... a little less definitive. It might be worthwhile to mention some of the non-breast cancer clinical studies in this space to add caution.
Response: As indicated by the reviewer, given the lack of clinical evidence for probiotics and prebiotics we have editing the language in these sections of the manuscript indicating that these could exert important anticarcinogenic effects in breast cancer that could indicate their employment as adjuvants in standard-of-care breast cancer treatments. In addition, we have previously mention in the manuscript the effect of probiotics in a clinical trial for colon cancer (reference 92), but now we have also included as reference a review about of the use of probiotics as adjuvants in anticancer therapy in other cancer types (reference 93).
-Comment: The data for the role of phytoestrogens and cancer risk versus cancer recurrence are variable and not high level. It would be worthwhile to be more inclusive of this uncertainty - ie https://pubmed.ncbi.nlm.nih.gov/18614351/
Response: As suggested by the reviewer, we have now included a paragraph describing several studies investigating the role of phytoestrogens and cancer risk versus cancer recurrence (Page 14, lines 434-437 and lines 444-459).
Comment: It would strengthen the conclusion to discuss top consistent findings from the review, which taxa come up more than once, any ongoing trials looking at immunotherapy efficacy based on microbiome in breast cancer, main gaps.
Response: Many thanks for the very interesting suggestion. We have strengthened the conclusion with the main gaps suggested by the reviewer (Page 17, lines 582-584 and page 18 lines 607-609).
-Comment: Some minor editing for English would strengthen it.
ie line 67, overweight or obesity should be corrected to being overweist or obese..
Response: This sentence has been changed in the revised manuscript.
Line 72 one example... is followed by two examples.
Response: This mistake has been corrected.
220, "data" are plural
Response: This mistake has been corrected
The legend for Figure 1 needs editing attention
Response: Figure 1 has been editing and redesigned.
Reviewer 2 Report
The aim of this review was to present the current state of knowledge on the role of the breast and gut microbiome in the pathogenesis of breast cancer. In recent years, more and more studies have been conducted on the composition of the microbiome, mainly the intestinal microbiome, and its impact on human health. In my opinion the subject matter of this review is topical and will be of interest to many readers.
The link between breast cancer and the composition of the microbiome results from the possibility, with the participation of bacterial beta-glucuronidase and/or beta-glucosidase of bacterial origin, to promote estrogen recirculation, to induce the synthesis of estrogen-inducible growth factors, as well as the regulation the immune response by promoting of Treg lymphocytes proliferation and through pathogen-induced chronic inflammation , and possibly by epigenetic reprogramming. These findings could give a new possibilities for the prevention and treatment of breast cancer.
It is a well written, logically arranged and up to date review. The references seem to be adequate. I have only a few suggestions:
- Line 72: Helicobacter pyroli should be written in italics.
- Lines 391 & 522: spelling of ‘β-glucuronidase’ to be corrected (β- instead of b- or beta-)
- Fig. 1 - too small font in the figure, words difficult to read at 100% magnification.
- Fig. 2 - too small font in the Probiotics box, a lot of strange characters - question marks in the rectangle between words
I recommend the manuscript for publication after minor revision.
Author Response
-Comment: The aim of this review was to present the current state of knowledge on the role of the breast and gut microbiome in the pathogenesis of breast cancer. In recent years, more and more studies have been conducted on the composition of the microbiome, mainly the intestinal microbiome, and its impact on human health. In my opinion the subject matter of this review is topical and will be of interest to many readers.
The link between breast cancer and the composition of the microbiome results from the possibility, with the participation of bacterial beta-glucuronidase and/or beta-glucosidase of bacterial origin, to promote estrogen recirculation, to induce the synthesis of estrogen-inducible growth factors, as well as the regulation the immune response by promoting of Treg lymphocytes proliferation and through pathogen-induced chronic inflammation , and possibly by epigenetic reprogramming. These findings could give a new possibility for the prevention and treatment of breast cancer.
It is a well written, logically arranged and up to date review. The references seem to be adequate. I have only a few suggestions:
- Line 72: Helicobacter pyroli should be written in italics.
- Lines 391 & 522: spelling of ‘β-glucuronidase’ to be corrected (β- instead of b- or beta-)
- Fig. 1 - too small font in the figure, words difficult to read at 100% magnification.
- Fig. 2 - too small font in the Probiotics box, a lot of strange characters - question marks in the rectangle between words
I recommend the manuscript for publication after minor revision.
Response: We thank the reviewer for their useful comments and suggestions. We have corrected the concerns pointed by the reviewer in the revised manuscript.